# Telomerase and CD4 T Cell Immunity in Cancer

**DOI:** 10.3390/cancers12061687

**Published:** 2020-06-25

**Authors:** Magalie Dosset, Andrea Castro, Hannah Carter, Maurizio Zanetti

**Affiliations:** 1The Laboratory of Immunology, Department of Medicine and Moores Cancer Center, University of California San Diego, 9500 Gilman Drive, La Jolla, CA 92093-081, USA; magalie.dosset@gmail.com; 2Division of Medical Genetics, Department of Medicine and Bioinformatics and Systems Biology Program, University of California San Diego, La Jolla, CA 92093, USA; andreabcastro@ucsd.edu (A.C.); hkcarter@health.ucsd.edu (H.C.); 3Health Science, Department of Biomedical Informatics, School of Medicine, University of California San Diego, La Jolla, CA 92093, USA

**Keywords:** telomerase, TERT, CD4 T cells, MHC-II, cancer, immune surveillance, immune monitoring, prognostic-predictive biomarker, vaccine, immune checkpoint therapy

## Abstract

Telomerase reverse transcriptase (TERT) is a conserved self-tumor antigen which is overexpressed in most tumors and plays a critical role in tumor formation and progression. As such, TERT is an antigen of great relevance to develop widely applicable immunotherapies. CD4 T cells play a major role in the anti-cancer response alone or with other effector cells such as CD8 T cells and NK cells. To date, efforts have been made to identify TERT peptides capable of stimulating CD4 T cells that are also able to bind diverse MHC-II alleles to ease immune status monitoring and immunotherapies. Here, we review the current status of TERT biology, TERT/MHC-II immunobiology, and past and current vaccine clinical trials. We propose that monitoring CD4 T cell immunity against TERT is a simple and direct way to assess immune surveillance in cancer patients and a new way to predict the response to immune checkpoint inhibitors (ICPi). Finally, we present the initial results of a systematic discovery of TERT peptides able to bind the most common HLA Class II alleles worldwide and show that the repertoire of MHC-II TERT peptides is wider than currently appreciated.

## 1. Introduction

T cells are at the core of “immune surveillance” and are the best candidates to control cancer cells in an antigen-specific manner [1,2]. CD8 T cells recognize ~9 amino acid long peptides associated with the MHC-I molecule and kill target cells displaying the corresponding MHC-I/peptide combination. In humans, tumor-specific CD8 T cells are present in patients with hematologic malignancies and solid tumors [3,4,5,6], often expressing the exhausted PD-1 phenotype. Therapeutic vaccines to induce CD8 T cell responses have been only partially successful [7,8]. CD4 T cells recognize ~15 amino acid long peptides associated with the MHC-II molecule. In cancer, CD4 T cells have received less attention than CD8 T cells because tumor cells have reduced, and often lacking, expression of MHC-II molecules [9,10] and because the Class II-associated invariant chain peptide (CLIP) prevents presentation of endogenous peptides [11].

Notwithstanding these issues, CD4 T cells play a central role in orchestrating the adaptive immune response through multiple functions. CD4 T cells can be directly cytotoxic but also cooperate with B cells [12], CD8 T cells [13], and CD4 T cells [14]. CD4 T cells also play a pivotal role in the generation and maintenance of memory CD8 T cells [15,16,17,18]. Finally, CD4 T cells can have suppressive (Tregs) [19] and inflammatory (Th17) [20] activity. Collectively, CD4 T cells form a class of T cells with functions often opposite to each other. The complex array of functions by CD4 T cells in relation to anti-tumor immunity has been reviewed recently [21,22].

The mechanisms of tumor protection by CD4 T cells stem from studies in mice. Based on adoptive transfer of tumor-reactive CD4 T cells or CD4 T cell depletion, it was initially demonstrated that CD4 T cells are necessary for protection against tumors lacking MHC-II [23,24,25,26,27,28,29]. These experiments showed that activated CD4 T cells induce delayed type hypersensitivity (DTH)-like reactions and attract inflammatory cells (macrophages, granulocytes, eosinophils, and NK cells) in or around the tumor [27,30]. Protection was linked with IFN-γ secretion, which in turn induces reactive oxygen species and nitric oxide, inhibits angiogenesis, and activates cytotoxic macrophages [28,31,32,33,34,35,36,37]. Subsequently, however, it became apparent that IFN-γ secreted by Th1 CD4 T cells causes the upregulation of MHC-II molecules on the surface of tumor cells, enabling MHC II-restricted killing [38,39,40]. Thus, CD4 T cells can kill tumor target cells (i) that constitutively express MHC-II molecules via direct MHC-II/peptide recognition [26,38,41,42] or (ii) indirectly by inducing MHC-II expression via IFN-γ. In either case, lysis of target cells occurs via the release of cytokines (IFNγ, TNFα, Perforin/Granzyme B) [39,40] or via interactions with Fas or TRAIL apoptosis-inducing receptors expressed on cancer cells [43,44].

In cancer patients, CD4 T cells recognize unmutated self-tumor antigens, viral antigens causative of tumor transformation, and mutant peptides (neoantigens) resulting from non-synonymous mutations or gene fusion in the cancer genome [22]. Because MHC-II peptides often show an average MHC binding affinity less stringent than that of MHC-I restricted peptides [45] and have more promiscuous MHC binding characteristics [46,47,48], CD4 T cells could have a wider range of regulation of the antitumor response, suggesting their relevant role in immune surveillance. In cancer patients, MHC-II restricted CD4 T cell responses against self-antigens have been detected in the circulation and at the tumor site [49,50,51,52,53,54,55,56,57,58], and a high density of tumor-infiltrating CD4 T cells correlates with good prognosis in many cancer types [59,60].

In this review, we discuss CD4 T cell responses against telomerase reverse transcriptase (TERT), a ubiquitous tumor antigen [61,62]. Specifically, we will review existing data on systemic TERT-specific CD4 T cell immunity as a biomarker of antitumor immunity and predictor of clinical outcome and present new data on the identification of new MHC-II TERT peptides with wide spectrum HLA alleles binding characteristics.

## 2. TERT and Cancer

The “two-hit” model states that a dominantly inherited predisposition to cancer requires a germline mutation, while tumorigenesis requires a second, somatic mutation. Non-hereditary cancers of the same type also require two hits, but both are somatic [63,64]. However, two hits only determine whether or not a somatic cell turns into a cancer cell, without guaranteeing that the cell will concomitantly acquire self-renewal properties. Activation of the holoenzyme telomerase enables cells that have accumulated at least two mutations to escape senescence and to enter self-renewal [65]. In 1999, Hahn formally demonstrated that the ectopic expression of the catalytic subunit of telomerase (TERT) in cells with SV40 large-T antigen and activated H-Ras resulted in direct tumorigenic conversion of normal human epithelial and fibroblast cells. These transformed cells were shown to form tumors in nude mice [66]. Thus, although telomerase per se is not tumorigenic, it plays a direct role in oncogenesis by allowing pre-cancerous cells to proliferate continuously and become immortal.

Telomerase is a ribonucleoprotein that mediates RNA-dependent synthesis of telomeric DNA, maintaining telomere length and chromosomal stability [67,68,69]. Approximately 85‒90% of all human tumors express high telomerase activity [70,71], while normal tissues display no or little activity [70,71]. Since telomeres shorten progressively with successive cell divisions. Telomerase is intimately linked with the tumorigenic process. However, while ~90% of human cancers depend on telomerase presence and activity, a small percentage (~10%) use an alternative lengthening of telomeres (ALT) mechanism [72].

In the past two decades, it has also become apparent that TERT is expressed at every stage of the cancer process, from the incipient cancer stem/tumor initiating cell through to the metastatic cancer cell [73,74], playing an essential role in each stage (Figure 1). For extended discussion, see [62].

Briefly, TERT is expressed in cancer stem cells and progenitor cells, where it is indispensable for self-renewal [75]. In the clinical condition dyskeratosis congenita, telomerase mutations that inactivate its enzymatic activity lead to bone marrow failure [76]. Indeed numerous reports show that cancer stem cells (CSC) depend on TERT for their ability to self-renew and for tumor propagation [77,78,79,80]. TERT is also expressed in circulating tumor cells (CTC) shed from the primary tumor [79,81,82] and is required for epithelial-mesenchymal transition (EMT) [83]. In addition, chemoresistant tumor cells can also upregulate TERT antigen [84]. Thus, it is not surprising that TERT expression levels in tumors correlate with poor prognosis in several cancer types [85,86] including lung [87] and breast [88,89] cancers.

Unlike most conserved self-tumor antigens, TERT expression is additionally regulated by mutations in the promoter region. The human TERT promoter lacks both TATA and CAAT boxes but is highly GC-rich. While it is inactive in normal and pre-immortal cells, it is de-repressed in cancer cells. The human TERT promoter contains binding sites for transcription factors such as c-Myc, Sp1, the human papilloma 16 E6 protein, and steroid hormones (estrogen and androgens), each contributing to positive expression regulation. In the past decade, it became apparent that in numerous cancer types the TERT promoter carries mutations. Remarkably, these are so frequent that TERT promoter mutations are the most frequent mutations in the cancer genome [90,91]. They preferentially involve mutually exclusive nucleotide changes such as -124C > T and -146C > T from the ATG start site and CC > TT tandem mutations at -124/-125 and -135/-139 from the ATG start, albeit the latter have lower frequency. The presence of TERT promoter mutations has been associated with increased TERT expression [92], cancer recurrence, and treatment resistance [93,94,95]. Cancers that are frequently associated with TERT promoter mutations include glioblastoma multiforme (GBM), melanoma, hepatocellular carcinoma, urothelial cancers, anaplastic thyroid cancer, and a variety of non-melanoma skin cancers (for review see [96]). Because TERT is expressed in >90% cancers in humans, affects cancer cells at every stage of cancer differentiation, and TERT promoter mutations are very frequent in the cancer genome, we argue that TERT remains an ideal conserved self-tumor antigen for immunological interventions to curb cancer cell growth and prolong patient survival [62]. 

## 3. TERT-Specific CD4 Th1 Cells as Pivotal Modulator of the Anti-Tumor Immune Response

### 3.1. Current MHC-II Restricted TERT Peptides 

In 2000, this laboratory provided the first evidence that TERT was immunogenic and could expand cytotoxic CD8 T cells in the peripheral blood of cancer patients [97]. For much of the next 10 years, efforts continued to be focused on CD8 T cells (reviewed in [61,98]). The identification of MHC-II TERT peptides came as a second wave [99,100,101,102,103,104,105,106,107,108] (Table 1). Peptides were identified through prediction software for their ability to bind multiple HLA Class II alleles commonly expressed in the Caucasian population (HLA-DR1, HLA-DR3, HLA-DR7, or HLA-DP4). Curiously, in some instances peptides also included a MHC-I binding sequence (e.g., the GV1001 peptide) and were therefore selected on the assumption that the concomitant stimulation of CD4 and CD8 T cells would elicit a more potent antitumor response [109]. 

The MHC-II TERT peptides identified to date are listed in Table 1. Although different methods to test immunogenicity do not allow for a reliable comparison, it is evident that these peptides differ with respect to degree of binding across diverse HLA haplotypes (immunoprevalence) and their ability to induce an anti-TERT response (immunodominance). For instance, it has been reported that HLA-DR alleles were more frequently involved in antitumor T cell immunity than HLA-DP4 [48,114,115]. In a study conducted in a cohort of 87 lung cancer patients, Laheurte et al. [107] previously compared the immunogenicity of a pool of four TERT peptides selected on the basis of a prevalent binding to HLA-DR (termed universal cancer peptides or UCP) to that of a pool of HLA-DP4 binding peptides (p613, p911, p573, p543). The results showed that the number of TERT specific IFNγ-secreting T cells after stimulation with the pool of HLA-DR peptides was generally two to three-fold higher than that of the HLA-DP peptide pool. Similar results were obtained in other cancers like melanoma, breast cancer, renal cell carcinoma, and colon cancer [107]. HLA-DR restricted peptides were also more useful at assessing pre-existing anti-TERT immunity in individual patients since the percentage of lung cancer patients responding to HLA-DR restricted peptides was greater than that of patients responding to HLA-DP (25% vs 10%, respectively) [107]. 

### 3.2. Prognostic Value of Systemic Anti-TERT CD4 T Cell Immunity in Cancer

Several studies demonstrated that the presence of T cells in the tumor microenvironment provides valuable insights on a patient’s clinical outcome and response to immune checkpoint inhibitors (ICPi) therapy [56,59,60]. However, little is known about the specificity and function of intratumor T cells in human cancer. Monitoring antitumor T cell immunity in blood is a simple, non-invasive alternative as blood is in between the primary activation site of T cells (lymph node) and their effector site (tumor), suggesting that systemic antitumor immunity may mirror what happens in the tumor [116,117]. Recent evidence indicates that antitumor T cell immunity in blood and intra-tumor T cell abundance predict clinical outcome [118,119,120]. Importantly, the role of systemic CD4 T cells has been emphasized in two recent studies conducted in lung cancer where the presence of functional CD4 Th1 cells in blood at the baseline (pre-existing immunity) proved to predict clinical response to anti-PD-1/PD-L1 immunotherapy [121,122].

We and others established the existence of CD4 Th1 cell specific for different TERT epitopes in several cancers including leukemia, lung, colon, melanoma, renal, and liver cancers (Table 2). A CD4 Th1 response against the pool of UCP peptides was detected before any treatment in about 25% of blood samples from metastatic non-small cell lung (NSCLC) [105,110], anal [111] or renal [112] cancer patients (Table 2). A similar proportion of TERT-responders at the baseline was reported in advanced melanoma against the single GV1001 peptide [103], while no response against this epitope was detected in metastatic NSCLC patients [123] (Table 2). As expected, the frequency of TERT Th1 responders against the UCP peptide pool was greater in localized vs metastatic NSCLC (45% vs. 24%, respectively) [110], pointing to a link between functional anti-TERT CD4 T cell immunity in peripheral blood and tumor progression. Interestingly, it was found that chemotherapy potentiates the protective effect of systemic anti-TERT Th1 immunity. Among NSCLC patients with controlled disease after platinum-based chemotherapy, the group with a positive UCP-specific Th1 response at the baseline had a three month extension in overall survival (OS) compared to TERT non-responders [105] (Table 2). By contrast, no benefit of the presence of a pre-existing anti-TERT immunity was observed in patients for whom the treatment failed to stabilize or reduce tumor burden. It appears as if therapies promoting immunogenic cell death [124,125] increase the effector function of pre-existing CD4 T cell immunity, hence resulting in improved survival [126]. Synergy between immunogenic cell death and anti-TERT may also be enhanced by removing immune suppression. In a preliminary study conducted in metastatic renal cell carcinoma, the transient depletion of immunosuppressive Tregs that occurs after rapalog everolimus was associated with heightened systemic anti-TERT Th1 cell responses, a 1.5 fold increase in individual TERT responders, and an improvement of progression-free survival (PFS) [112] (Table 2). In advanced anal cancer patients, an increase in the magnitude of anti-TERT Th1 responses was observed after immunogenic chemotherapy by docetaxel, cisplatin, and 5-fluorouracil and greater progression free survival [111]. A plausible hypothesis is that chemotherapy-induced immunogenic cell death increases the efficiency of CD4 T cell activation [126,127]. Alternatively, chemotherapy-induced cell death could cause the release of tumor antigens and cause the activation of CD4 T cells specific for these antigens. Interestingly, an oxaliplatin-resistant colorectal cancer cell line was shown to express increased TERT levels [84]; this suggests that the input of another effective cytotoxic drug to manage chemoresistant tumor cells may facilitate anti-TERT immunity and possibly activate novel CD4 T cell clones.

Although the mere presence of pre-existing systemic anti-TERT CD4 T cells was not sufficient to predict survival in NSCLC patients [105], greater baseline values correlated with stronger protection, both in metastatic and localized NSCLC after chemotherapy (median OS of 17 vs. 9 months in anti-TERT Th1^high^ vs anti-TERT Th1^low^, *p* = 0.023) [110]. This confirms that systemic anti-TERT CD4 T cells are important and their expansion after treatment is critical for a durable control of disease progression. Similarly, a study by Voutsas et al. [128] showed that a high level of HER-2/neu-specific CD4 Th1 cells in peripheral blood pre-vaccination was associated with a more favorable outcome. It remains to be determined whether these effects also reflect clonal diversity even though CD4 (but not CD8) T cell clonal diversity prior to CTLA-4 blockade significantly improved survival in melanoma patients [129].

The percentage of patients responding to TERT at baseline was found to correlate inversely with disease stage [110]. Since TERT antigen expression tends to increase with disease progression [73,74], a drop in TERT responders in metastatic patients may be attributed to immunosuppression. For instance, in vitro studies show that removal of myeloid derived suppressor cells (MDSC) [130] and PD-1/Tim-3 blockade [110] increases TERT-specific CD4 Th1 cell response in certain patients. This is consistent with recent reports showing that peripheral CD4 T cells positively influence the outcome of immune checkpoint blockade [121]; in addition, a high level of functional systemic CD4 Th1 cells prior to anti-PD-1 therapy correlates with increased PD-1^+^ CD8 T cells and better survival [122], and a diversified pre-existing blood CD4 T cell repertoire predicts better clinical outcome to CTLA-4 blockade [129]. Therefore, enhancement of the TERT response by peripheral CD4 T cells in vitro by immune checkpoint inhibiting antibodies could represent a valuable tool to predict the in vivo response to ICPi. In support of this idea is a recent study showing that the clonality of tumor-infiltrating T cells after PD-1 blockade dramatically differs from that of tumor-infiltrating T cell clonotypes identified at baseline in patients with basal or squamous cell carcinoma [131]. This suggests that immune checkpoint inhibitors also act by recruiting peripheral T cells in addition to reinvigorating pre-existing tumor-infiltrating lymphocytes. Importantly, NSCLC patients with increased systemic anti-TERT CD4 T cell immunity after anti-PD-1 therapy were shown to have a better outcome [132]. Altogether, monitoring of anti-TERT CD4 T cell responses in vitro could greatly help refine the stratification of cancer patients and predict clinical outcome in response to immune checkpoint blockade (Figure 2).

### 3.3. Past and Current Therapeutic Approaches Targeting Anti-TERT CD4 Th1 Cell Immunity

The therapeutic efficacy of the first identified MHC-II TERT peptides has been evaluated in cancer patients. All past therapeutic TERT-vaccine trials have been recently reviewed [62]. GV1001 is currently the sole reported MHC-II peptide shown to induce TERT-specific CD4-T cells in a clinical trial setting. GV1001 is a peptide vaccine representing a 16 amino acid TERT sequence that binds multiple MHC-II molecules and also contains putative MHC-I epitopes. GV1001 vaccination was evaluated in different cancer types either alone or in combination with MHC-I (HLA-A2.1) TERT peptides (p540) in granulocyte macrophage-colony stimulating factor (GM-CSF) adjuvant, with or without chemotherapy [62]. Administered alone, GV1001 was poorly immunogenic since it induced a response in only 17% (1/6) of patients with cutaneous T cell lymphoma [133]. In contrast, when administered in combination with GM-CSF, with or without chemotherapy, the immune response rate generally varied between 50% to 80% irrespective of cancer type [103,123,134,135]. In pancreatic cancer, ~65% of patients had a specific CD4 Th1 cell response after GV1001 + GM-CSF, with or without concurrent gemcitabine [134,135]. The response elicited by GV1001 + GM-CSF was associated with a clear improvement of overall survival (median OS 7 vs. 3 months in TERT responders and non-responders respectively, *p* = 0.0001) [134]. No overall survival data are available for the combination GV1001+GM-CSF gemcitabine or gemcitabine + capecitabine. An immune response rate of ≥70% was observed in melanoma patients [103]. In NSCLC, 54% of subjects had a specific response after GV1001+p540+GM-CSF vaccine with improved survival (median OS 19 vs 3.5 months in TERT responders and non-responders, *p* < 0.001) [101,123]. Interestingly, the use of docetaxel and post-operative chemotherapy with radiation therapy in combination with GV1001 in NSCLC increased the immune response rate to 80%, confirming the benefit of immunogenic cell death-inducing drugs to boost vaccine efficacy [123]. Finally, GX301 vaccine, which contains a mix of MHC-II peptides (GV1001, p672, p711) + p540, was shown to activate CD4 T cell responses in 64% of patients with prostate and renal cancer, leading to a drastic improvement in overall survival (median OS not reach in TERT responders vs. 3.3 months, *p* = 0.0002). Collectively, this shows the importance of stimulating a diverse, polyclonal CD4 T cell response for heightened vaccine efficacy. 

More recently, a Phase I study based on a DNA construct which encodes for an inactive TERT fused to human ubiquitin (INVAC-1) [136] showed that patients with advanced solid tumors efficiently generate both TERT-specific CD4 T cells and cytotoxic CD8 T cells associated with a dramatic decrease (67%) of systemic regulatory T cells. Clinically, 58% (15/26) of patients had stable disease with a median overall survival of 15 months; 65% of subjects were still alive after 1 year [137,138]. 

Ongoing clinical trials that evaluate novel TERT MHC-II-based vaccines alone or combined with ICPi are listed in Table 3.

Based on the foregoing, one notes that MHC-II-based TERT vaccines generally outperformed the immunological and therapeutic efficacy of MHC-I TERT vaccines [62]. It also appears that the use of multiple promiscuous MHC-II TERT peptides [139] may be a useful strategy to diversify the clonality of responding CD4 T cells. Finally, attention should be paid to the class of T cells induced by vaccination. A proposal would be to target the induction of TERT-specific memory T cells with stem-like characteristics (T_SCM_) [140,141] and tumor-resident memory T cells (T_RM_) [142,143], two subsets with strong antitumor properties that were reported to be involved in the response to ICPi [144,145,146,147].

## 4. Prospects for the Identification of Novel Immunogenic TERT CD4 Epitopes

Until now, only a handful MHC-II-restricted TERT peptides have been identified and used implying that a systematic discovery may lead to the identification of additional MHC-II peptides with immunogenic properties. Here, we present our initial results on the identification of additional promiscuous peptides across multiple HLA alleles. To this end, we evaluated NetMHCIIpan3.2 [148] affinity predictions for all supported MHC-II alleles commonly found (≥10%) worldwide (Figure 3) for all unique 15mer peptides derived from four TERT transcripts including the canonical full length transcript (1132aa) and three alternatively spliced variants (α-deletion, β-deletion, γ-deletion) [149]. 

While alternatively spliced transcripts have been frequently studied in the context of TERT function in different tissues and different developmental stages [150], unique peptides derived from novel splice junctions or skipped exons have not been taken into consideration, particularly with respect to immunogenicity. Consequently, here we incorporated alternatively spliced transcripts in our analysis. From the canonical full-length and three alternatively spliced transcripts, we identified 1190 unique peptides (Figure 4a) (median affinity ranging from 1.9 to 95 percentile rate) and retained the top 30 most promiscuous, i.e., those able to bind a large fraction of MHC-II alleles (Figure 4b). While the majority of the top 30 peptides were shared between all examined transcripts, we observed novel peptides not seen in the canonically studied transcript. Interestingly, some predicted peptides are completely novel, while others contain sequences previously identified as MHC-I epitopes. Most of the top 30 peptides are predicted to bind >60% of the 1911 most common MHC-II alleles (Figure 4b).

Interestingly, the prediction revealed that only two of the currently known MHC-II TERT peptides, UCP1 and partially p541, were among the predicted top 30 (2.5%) 15mers, binding to 82% and 80.5% of common HLA Class II alleles, respectively (Figure 4b). Other currently studied TERT peptides p660, p672, and p673 ranked in the top 10% of total 15mers with median affinity scores close to the maximum binding threshold of 10. The UCP2, UCP3, UCP4, GV1001, p68, p663, and p766 peptides were predicted to have poorer binding properties (Figure 5). Thus, this analysis confirms our expectation that the repertoire of MHC-II TERT peptides that are potentially immunogenic in a large segment of the population is wide, and certainly wider than the one identified to date.

## 5. Conclusions

We have provided herein a comprehensive view of the role of CD4 T cell immunity against TERT in cancer. Because of its expression during all stages of tumor differentiation, TERT remains an important immunological target for immunotherapy. The main points we covered can be summarized as follows. Based on a handful of identified and validated peptides, it appears clear that cancer patients often display CD4 T cell reactivity against TERT. Notably, the expansion of precursor CD4 T cells in vitro indicates that these cells have not been deleted in the thymus, have survived shaping of the repertoire over the immunological history of the individual, and persist as part of the available T cell repertoire. The expansion of TERT-specific CD4 T cells in the peripheral blood of cancer patients correlates with a more favorable outcome of disease. For this reason, we propose that a systematic assessment of CD4 T cell immunity to TERT in circulating lymphocytes could be used to predict the response to immune checkpoint inhibitors, even though the ultimate anti-tumor effector function may be against tumor antigens other than TERT, for example neoantigens. TERT-specific CD4 T cells would nevertheless enable other responses by providing an initial attack on tumor cells and/or helping the expansion of CD4 and CD8 T cells with different tumor antigen specificities (epitope spreading).

We also show that the number of potential MHC-II binding TERT peptides far exceeds the number of peptides presently known. Additional binders with a broad MHC-II spectrum have been identified including some unique to TERT splice variants. At this point, whether or not these new peptides are also immunogenic in vivo will need to be determined experimentally. Moreover, it is important to establish experimentally if MHC-II peptides with the best affinity scores are also the most immunogenic. Since the predictions made in this report are for peptides with a large coverage of MHC-II alleles in the human population, experimental validation will prove of great relevance to better understand the role of TERT CD4 T cell immunity in immune surveillance and for immunotherapy of cancer.

## Figures and Tables

**Figure 1 cancers-12-01687-f001:**
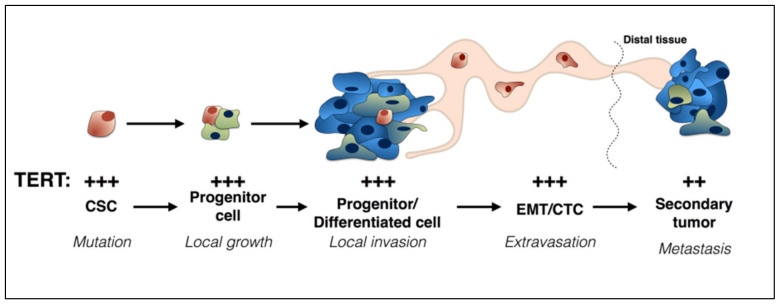
Telomerase reverse transcriptase (TERT) is expressed at every stage of cancer progression. The sign + refers to gene transcription.

**Figure 2 cancers-12-01687-f002:**
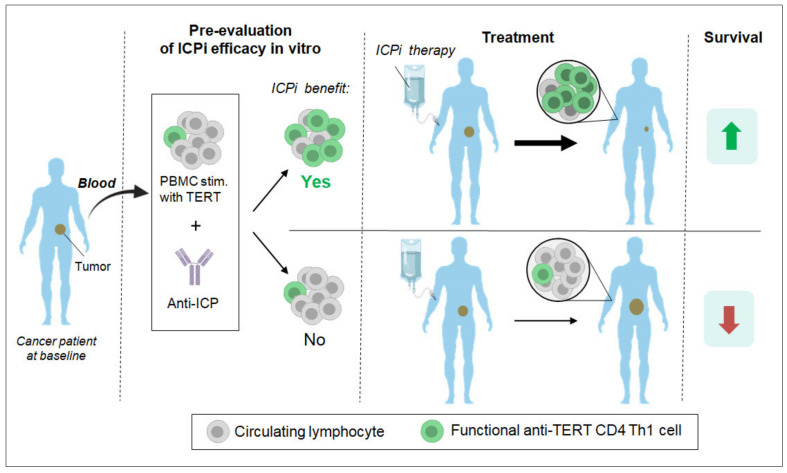
Proposed strategy to identify cancer patients most likely to respond to immune checkpoint inhibitors (ICPi) therapy. We propose to select patients for ICPi therapy based on an in vitro stimulation experiment evaluating the capacity of ICP blockade to stimulate systemic anti-TERT CD4 T cell immunity. Peripheral blood mononuclear cells (PBMC) from patients collected at the baseline would be stimulated with MHC-II TERT peptides in the presence of anti-ICP antibodies. Since anti-TERT Th1 immunity was generally associated with a good prognosis [110,111,112], a drastic increase of anti-TERT response following ICP blockade in vitro would ensure that the patient can benefit from the cognate ICPi therapy.

**Figure 3 cancers-12-01687-f003:**
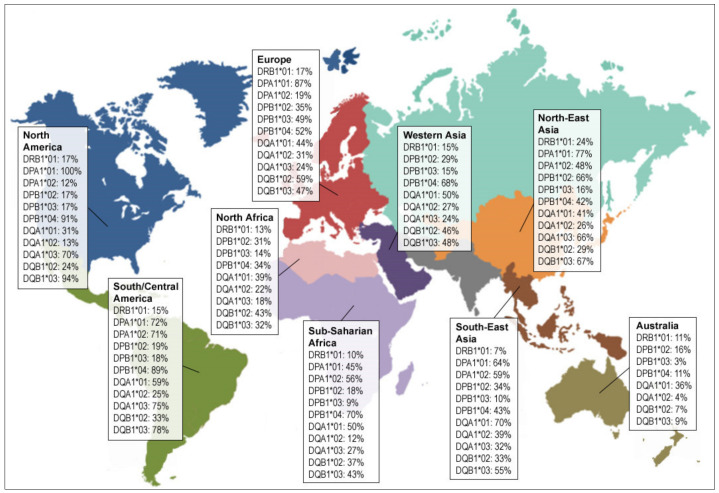
Prevalence of the most common (≥10% frequency) MHC-II molecules worldwide (according to Allele Frequency Net Database, http://www.allelefrequencies.net). All class II loci (DRB1, DQA1, DQB1, DPA1, DPB1) were considered in each region available. Alleles with minimum 2 field resolution, ≥0.1 allele frequency, and presence in >50% of regions were retained. Any α/β molecule containing at least one retained common allele was included, resulting in 1911 studied molecules.

**Figure 4 cancers-12-01687-f004:**
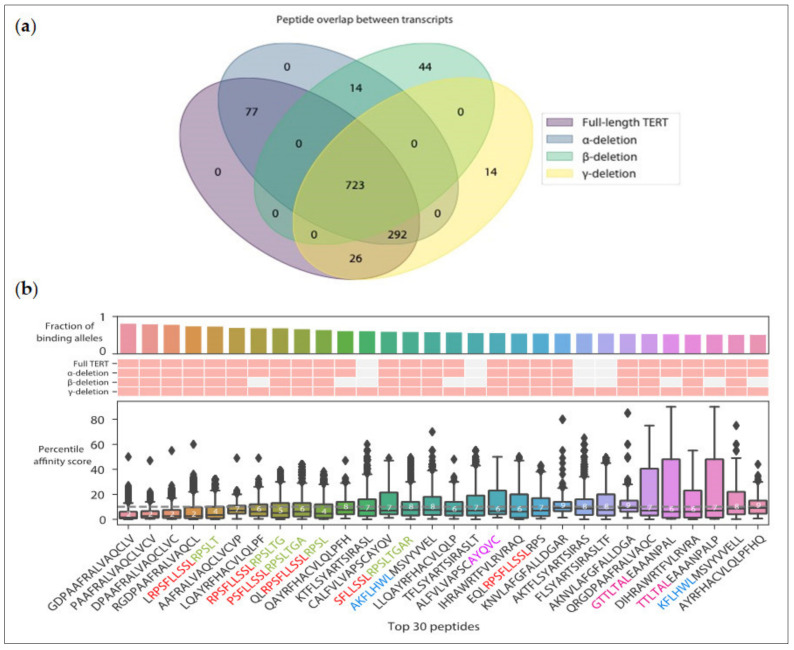
Prediction of the top 30 promiscuous TERT peptides. Predicted affinity scores of 15mer TERT peptides across 1911 molecules containing common MHC-II alleles (>10% expression, see Figure 3) were obtained using NetMHCIIpan3.2. (**a**) Venn diagram showing the overlap of unique 15mers originating from 4 TERT transcripts (full length, minus-α, minus-β, and minus-γ TERT transcripts). (**b**) Top 30 predicted promiscuous TERT MHC-II peptides are shown. Peptides with an affinity score <10 are considered binders [148]. (Top) Barplot indicating the fraction of common MHC-II alleles that can be bound to each peptide. (Middle) Heatmap with red boxes indicating each peptide’s transcript of origin. (Bottom) Boxplots denoting the distribution of affinity scores for all unique 15mer peptides from each TERT transcript studied. Known MHC-I binding motifs are highlighted and identified using different colors (red, green, blue, and purple) in respective 15mer sequences: R342 (red), R351 (green), I540 (blue), A167 (purple), L1107 (pink).

**Figure 5 cancers-12-01687-f005:**
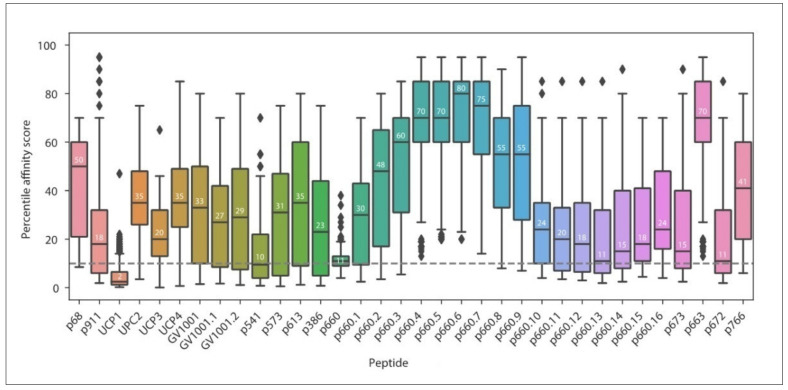
Overview of the promiscuous property of known TERT MHC-II-restricted peptides using NetMHCIIpan3.2. Predicted affinity scores for each known TERT peptide (Table 1) across 1911 molecules containing common MHC-II alleles (>10% expression, see Figure 3). Peptides with an affinity score <10 are considered binders. Affinity for peptides longer than 15 amino acids was evaluated for each 15mer within the sequence (sub-peptides indicated with decimal points). The median percentile rank affinity score is shown in each boxplot. The dotted line indicates the threshold for high affinity MHC binders.

**Table 1 cancers-12-01687-t001:** List of identified immunogenic TERT CD4 epitopes.

Peptide	Position	Sequence	Main HLA Restriction	Year	Ref.
p68	TERT_68-82_	APSFRQVSCLKELVA	HLA-DR	2018	[108]
p911 ^1^	TERT_911-927_	DEALGGTAFVQMPAH	HLA-DP4	2016	[107]
UCP1	TERT_44-58_	PAAFRALVAQCLVCV	HLA-DR	2012	[105,113]
UCP2	TERT_578-592_	KSVWSKLQSIGIRQH	HLA-DR		
UCP3 ^1^	TERT_916-930_	GTAFVQMPAHGLFPW	HLA-DR		
UCP4	TERT_1041-1055_	SLCYSILKAKNAGMS	HLA-DR		
p541	TERT_541-555_	LAKFLHWLMSVYVVE	HLA-DP4	2011	[102,103]
p573	TERT_573-587_	LFFYRKSVWSKLQSI	HLA-DP4		
p613 ^2^	TERT_613-627_	RPALLTSRLRFIPKP	HLA-DP4		
p386	TERT_386-400_	YWQMRPLFLELLGNH	HLA-DP4	2011	[104]
p660 ^3^	TERT_660-689_	ALFSVLNYERARRPGLLGASVLGLDDIHRA	HLA-DR		
p663^3^	TERT_663-677_	SVLNYERARRPGLLG	HLA-DR		
p673 ^3^	TERT_673-687_	PGLLGASVLGLDDIH	HLA-DR		
GV1001 ^2^	TERT_611–626_	EARPALLTSRLRFIPK	HLA-DP4	2006	[101]
p766	TERT_766-780_	LTDLQPYMRQFVAHL	HLA-DR	2003	[100]
p672^3^	TERT_672-686_	RPGLLGASVLGLDDI	HLA-DR	2002	[99]

^1^, ^2^, ^3^ overlapping peptides.

**Table 2 cancers-12-01687-t002:** Relationship between TERT-reactive CD4 Th1 cells and patients’ survival.

Cancer type	Treatment	Responders		Overall Survival and anti-TERT CD4 T Cell Response at:		Ref.
		Baseline	Post-Treatment	Baseline	Post-Treatment	
Metastatic non-small cell lung cancer (NSCLC)	Platinum-based chemo therapies	38% (32/84)	ND	Patients with CD: median OS 53 months in TERT responders vs. 40 months in non-responders (*p* = 0.049)	-	[105]
Non-small cell lung cancer (NSCLC)	Platinum-based chemo therapies	45% (39/87) of localized 24% (20/83) of metastatic	ND	Two-year OS rate of 59% in anti-TERT Th1^high^vs. 22% in anti-TERT Th1^low^ (*p* = 0.006). Similar significant differences in localized and metastatic disease analyzed separately	-	[110]
Metastatic Renal cell carcinoma (mRCC)	Rapalog everolimus	48% (11/23)	74% (17/23) two months after treatment	ND	Better PFS achieved in patients with increased anti-TERT Th1 immunity and reduced Treg	[112]
Metastatic anal squamous cell carcinoma	Docetaxel, cisplatin and fluorouracil (DCF)	27% (17/64)	32% (16/50) one month after the last DCF cycle	Median PFS *p* = 0.059)	One-year PFS rate of 62.5% in TERT responders vs. 23.5 % in non-responders, (*p* = 0.017)	[111]

CD, controlled disease; OS, overall survival; PFS, progression-free survival; ND, not determined.

**Table 3 cancers-12-01687-t003:** Ongoing clinical trials based on the stimulation of TERT-specific CD4 Th1 cells.

TERT MHC-II Based Therapy	Cancer	Phase	Estimated Enrollment	Status	ID
UCPVax: pool UCPs peptides	Metastatic NSCLC	I/II	54	Recruiting	NCT02818426
UCPVax -Glio: pool UCPs peptides	Glioblastoma	I/II	28	Recruiting	NCT04280848
Optim-UCPVax: pool UCPs + Nivolumab (anti-PD-1)	Advanced NSCLC	II	111	Not yet recruiting	NCT04263051
VolATIL: pool UCPs (UCPVax) + Atezolizumab (anti-PD-L1)	Squamous Cell Carcinoma Cervical cancer Advanced Anal Carcinoma	II	47	Not yet recruiting	NCT03946358
GV1001 + Gemcitabine + Capecitabine	Pancreatic cancer	III	148	Unknown	NCT02854072
INVAC-1: modified TERT DNA plasmid	Chronic Lymphocytic Leukemia	II	90	Recruiting	NCT03265717

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
