# Peer review of "Telomerase and CD4 T Cell Immunity in Cancer"

_cancers, 2020, doi:10.3390/cancers12061687_

Round 1

Reviewer 1 Report

In this review, the authors presented a comprehensive introduction to the concepts of TERT/MHC-II immunobiology, and reviewed the current status of various clinical trials of TERT vaccines.   In addition, the authors posit that surveillance of CD4-cells immunity against TERT self-antigens would have clinical utility as therapeutic markers for responsiveness to immune-checkpoint agents, and provide new bioinformatics data on the diversity of immunogenic TERT peptides (MHC-II, CD4 selective), including previously unknown peptide sequences that correspond to various alternate spliced TERT isoforms. 

Overall, the authors did an excellent job introducing the salient aspects and therapeutic applications of CD4 immunity against TERT.  The review article is very well-written, and the discussion topics are logically arranged. Current and past clinical trials of TERT vaccines, including the most advanced (TERT peptide GV1001) in clinical development, produced mixed results so far. Disappointing data coming from the landmark TeloVac trial in 2014, which combines TERT vaccine with chemotherapy in advanced Prostate Cancers, were contrasted with recent success of GV1001 significantly reducing the clinical burdens of benign prostate hypertrophy.  These seemingly contradicting data were nicely resolved by the authors’ model of “a link between functional anti-TERT CD4 T cell immunity in peripheral blood and tumor progression”, suggesting that treatments with GV1001 will likely have marginal effects when tumor progression led to the development of an immunosuppressive environment.  Most importantly, the proposition to use surveillance of CD4-cells immunity against TERT self-antigens as therapeutic markers for checkpoint therapy is a testable hypothesis with immediate translation potential.  This alone warrants publication to encourage the clinical development of such applications in various types of cancers.  Some editorial and presentation changes/suggestions were listed at the end of this review to help clarify the key concepts.

In addition, I have a few questions for the authors, mostly stemming from my own curiosity for the topics being discussed, and by no means reflective of any omission on the authors’ part.  These are listed below:

1) The prevalence of TERT reactive CD4 cells are relatively high (~40%) in lung and kidney cancers (Table 2), but are there cancer type specific differences?  Specifically, is the increase in immunogenic signal against TERT a passenger effect of high immunogenicity in the so-called “hot” tumors?

2) Similarly, CD4-cells reactive to TERT self-antigens are postulated to escape negative selection at the thymus.  Is this escape universal?  Are there inter-individual differences in the ability to preserve these reactivities?

3) The authors suggested that cytotoxic chemotherapy may induce immunogenic cell death, thereby increasing the effector function of pre-existing CD4 T cell immunity against TERT.  However, another interpretation could be that following chemotherapy-induced tumor cytotoxic response, the immediate release of tumor antigens could promote direct, or B-cell mediated CD4 T-cell stimulation. Could this be possible?

4) While the total number of TERT alternatively spliced forms are high (~20), many of these isoforms are rarely expressed (except perhaps the alpha- and beta-delete isoforms). Given that many existing immunogenic TERT peptides are common to the different TERT isoforms, would the marginal increase in the quantification of TERT-immunogenicity warrants the additional costs for including the spliced-form-specific rare peptides for surveillance?

Minor/Editorial Points:

  • (line 81, extra ‘s’) two hits only determines
  • (line 83, double ‘of the’) activation of the of the
  • (line 142, missing ‘s’) different methods
  • (line 36, missing a comma) In humans, tumor-specific CD8 T cells are present in patients with hematologic malignancies and solid tumors [3–6], often expressing the exhausted PD-1 phenotype.:
  • (line 219) …in addition, high level of functional systemic CD4 Th1 cells prior anti-PD-1 therapy correlates with increased PD-1+ CD8 T cells and better survival reinforce this statement (likely an editorial comment not removed during proof-read)
  • The bottom of page 6 and the entire page 7 are not properly formatted

Suggestions for Figures:

  • Tables Formatting needs to be revisited
  • Figure 1 needs a legend, explaining how the cells are color-coded and what +++ and ++ refer to (mRNA/protein expression/activity level?)
  • Figure 4 also needs a legend explaining the color coding of the binding motifs

Author Response

Reviewer 1 1) The prevalence of TERT reactive CD4 cells are relatively high (~40%) in lung and kidney cancers (Table 2), but are there cancer type specific differences? Specifically, is the increase in immunogenic signal against TERT a passenger effect of high immunogenicity in the so-called “hot” tumors? Indeed, we believe that tumors owe their immunogenicity to the presence of an immune-inflamed tumor microenvironment. The fact that the highest frequency (~50%) of TERT responders was detected in metastatic renal cancers, a type of cancer considered immunogenic, and the lowest in metastatic anal carcinoma (~30%), supports the view. Regrettably, a correlation between circulating TERT-reactive CD4 T cells and the status of the tumor microenvironment was not assessed. Future studies will need to determine whether patients with no/low anti-TERT CD4 T cell immunity in the blood are those with poor T cell infiltration and high level local immunosuppression. 2) Similarly, CD4-cells reactive to TERT self-antigens are postulated to escape negative selection at the thymus. Is this escape universal? Are there inter-individual differences in the ability to preserve these reactivities? Although many self-reactive T cells are deleted in the thymus, deletion is not perfect. A recent work by Mark Davis suggested that “Clonal Deletion Prunes, but does not Eliminate Self-Specific αβ CD8+ T Lymphocytes” [Immunity. 2015 May 19; 42(5): 929–941]. We believe that a similar phenomenon may occur for CD4 T cells. Furthermore, since production of naïve T cells is maintained at some level throughout life, then change in T cell repertoire may occur. This argues that the available T cell repertoire in the adult individual harbors specificities against self-antigens including TERT. 3) The authors suggested that cytotoxic chemotherapy may induce immunogenic cell death, thereby increasing the effector function of pre-existing CD4 T cell immunity against TERT. However, another interpretation could be that following chemotherapy-induced tumor cytotoxic response, the immediate release of tumor antigens could promote direct, or B-cell mediated CD4 T-cell stimulation. Could this be possible? We agree with this reviewer and we have amended the text accordingly (page 5, line 196). 4) While the total number of TERT alternatively spliced forms are high (~20), many of these isoforms are rarely expressed (except perhaps the alpha- and beta-delete isoforms). Given that many existing immunogenic TERT peptides are common to the different TERT isoforms, would the marginal increase in the quantification of TERT-immunogenicity warrants the additional costs for including the spliced-form-specific rare peptides for surveillance? Although it is impossible to know a priori, it cannot be excluded that these alternative spliced variants are not shared with canonical TERT and may in fact be more immunogenic. These variants (especially beta and alpha deletion) co-exist with the canonical TERT mainly to regulate its activity and variation of their expression has been observed in different tumor types along cancer progression. A study reported the existence of TERT alternative variants in TERT-negative cells (Hrdličková R, Mol Cell Bio, 2012). From this observation we conclude that these variants may be expressed in tumors using ALT as a method to elongate telomeres and their peptides may be targetable in TERT-negative tumor cells. Minor/Editorial Points They have all been fixed. Suggestions for Figures Tables Formatting needs to be revisited Done Figure 1 needs a legend, explaining how the cells are color-coded and what +++ and ++ refer to (mRNA/protein expression/activity level?) Done Figure 4 also needs a legend explaining the color coding of the binding motifs Done

Reviewer 2 Report

The review is written clearly. Authors describe valuable information concerning overexpression of TERT and immune response via CD4 cells.  Their manuscript is concise and comprehensible for wide audience, but it contains all relevant information. I have found no serious flaws.

Author Response

Nothing to report

Reviewer 3 Report

This is an excellent review on the topic of telomerase and immunity that summarises the state of the art.

There are just a few minor things which have to be addressed by the authors including some minor grammar and language issues and some partly incorrect statements. 1. line 85: The paper from Hahn et al., 1999 describes 2 different scenarios depending on the order of events. Only when genomic instability by viruses such as SV40 or similar was introduces prior to hTERT expression plus activated ras or similar, tumour cells were induced. In contrast, when hTERT was the initial event the cells maintained a stable genotype and only had an extended lifespan but no cancer features. Please correct this statement since it is important to cite the paper correctly. 2. Line 95: Telomeres do not only shorten due to cell division and are therefore not a "mirror of the replicative history" but their length can be modified by changes of oxidative stress in the environment (von Zglinicki et al., 1995), von Zglinicki, 2000, 2002 etc). These statements need to be corrected.

Another question I have regarding the statement in line 30: do peptides against non-functional  TERT splice variants are functional in terms of anti-tumourigenicity?

The language issues are

a surplus "of" in line 83, line 116 HPV 16 E6 protein (space required), line 142: methodS, please explain what UCP means since it can also mean "uncoupling protein" which most likely is not meant here. line 196: replace "efficiently" with "efficiency", lines 266-268 do not make sense after "to trigger", please correct., line 273: ...had A stable disease...

Author Response

Reviewer 3

  • The paper from Hahn et al., 1999 ……. Please correct this statement.

The statement has been corrected

  • Telomeres do not only shorten due to cell division and are therefore not a "mirror of the replicative history"

The statement has been deleted.

  • do peptides against non-functional  TERT splice variants are functional in terms of anti-tumourigenicity?

To the best of our knowledge the immunogenicity of TERT spliced variants has not been studied. However, since immunogenic peptides do not overlap with the catalytic region, peptides against non-functional variants should also be able to induce an anti-TERT/ antitumor response.

Minor/Editorial Points

They have all been fixed.